# Risk Factors, Clinical Characteristics, and Antibiotic Susceptibility Patterns of *Streptococcal* Keratitis: An 18-Year Retrospective Study from a Tertiary Hospital in China

**DOI:** 10.3390/antibiotics13121190

**Published:** 2024-12-06

**Authors:** Zhen Cheng, Qingquan Shi, Bo Peng, Zijun Zhang, Zhenyu Wei, Zhiqun Wang, Yang Zhang, Kexin Chen, Xizhan Xu, Xinxin Lu, Qingfeng Liang

**Affiliations:** Beijing Institute of Ophthalmology, Beijing Tongren Eye Center, Beijing Tongren Hospital, Capital Medical University, Beijing 100005, China; 1901156@mail.ccmu.edu.cn (Z.C.); qingquan@mail.ccmu.edu.cn (Q.S.); christianpeng@mail.ccmu.edu.cn (B.P.); shenyu@ccmu.edu.cn (Z.Z.); weizhenyu@ccmu.edu.cn (Z.W.); eyewzq@163.com (Z.W.); biozy1@126.com (Y.Z.); ckxwl1234@hotmail.com (K.C.); xuxz0924@mail.ccmu.edu.cn (X.X.); luxinxin2009@126.com (X.L.)

**Keywords:** *Streptococcal* keratitis, bacterial keratitis, eye infection, antibiotic resistance

## Abstract

**Purpose:** Our aim was to investigate risk factors, clinical characteristics, and antibiotic susceptibility patterns of cornea-isolated *Streptococcus* species collected at a tertiary hospital in China over 18 years. **Methods:** This retrospective study reviewed data from 350 patients diagnosed with *Streptococcal* keratitis at Beijing Tongren Hospital between January 2006 and December 2023, including demographics, risk factors, clinical signs, in vivo confocal microscopy (IVCM) imaging, and antibiotic susceptibility testing. **Results:** The predominant type was *Streptococcus pneumoniae* (n = 108, 29.8%), followed by *Streptococcus mitis* (n = 90, 24.9%) and *Streptococcus oralis* (n = 85, 23.5%). Main risk factors included previous ocular surface disease (24.6%), ocular surgery (21.4%), and trauma (16.3%). Significant differences in clinical characteristics were observed among *S. pneumoniae*, *S. oralis*, and *S. mitis* regarding infiltration location (*p* = 0.038) and size (*p* = 0.037), as well as hypopyon presence (*p* = 0.006). IVCM revealed deeper inflammatory cell distribution and structural disruption as the disease progressed. Resistance rates of aminoglycosides, β-lactams, and fluoroquinolones have increased, with significant differences among species for amikacin (*p* = 0.010), gentamicin (*p* = 0.007), and others. Poor outcomes correlated with disease duration over one month, central corneal ulcers, dense infiltrations, hypopyon, and scar tissue presence on IVCM. **Conclusions:**
*Streptococcal* keratitis is a complex ocular infection with multiple risk factors. *S. pneumoniae*, *S. mitis*, and *S. oralis* are the primary causative agents, exhibiting varying clinical features and antibiotic resistance patterns. Key factors associated with poor outcomes include long disease duration, central corneal ulcers, and severe infiltration.

## 1. Introduction

Microbial keratitis (MK) is a serious, sight-threatening ophthalmological condition characterized by symptoms such as pain, redness, inflammation, and corneal opacity. It is the fourth leading cause of blindness worldwide, with 1.5–2.0 million new cases occurring yearly [1,2]. As one of the top three pathogens responsible for bacterial keratitis, *Streptococcal* keratitis typically presents with a deep, round central stromal ulcer with a progressive edge and anterior chamber hypopyon, and can result in corneal perforation, severe endophthalmitis, and vision loss [3,4,5,6,7].

*Streptococcal* keratitis is attributed to three distinct species of *Streptococcus*: *S. pneumoniae*, *S. mitis*, and *S. oralis*. Among these, *S. pneumoniae* is the most prevalent and is characterized by its high virulence, which can result in rapid disease progression and corneal perforation [8]. In contrast, *S. mitis* and *S. oralis* exhibit lower virulence, generally leading to milder clinical presentations [9]. Despite the significance of these pathogens, there is a paucity of studies addressing the clinical features and risk factors associated with *Streptococcal* species isolated from corneas, particularly with respect to the differences among infections caused by these three specific *Streptococcus* species. Furthermore, the treatment regimens and prognoses for these infections are not uniform, underscoring the necessity for comprehensive research in this area.

This retrospective study aimed to investigate the risk factors, clinical characteristics, and antibiotic susceptibility patterns of *Streptococcal* keratitis over an 18-year period. Additionally, we evaluated the clinical manifestations and antibiotic resistance rates of different species of cornea-isolated *Streptococcus*.

## 2. Results

### 2.1. Demographics and Predisposing Factors for Streptococcal Keratitis

In this retrospective study, a total of 350 patients diagnosed with *Streptococcal* keratitis were included. The demographic data are summarized in Table 1, indicating that 47.1% of the patients were women, and 56.0% of the cases involved the right eye. The participants’ ages ranged from 2 months to 89 years, with a mean age of 49.1 years (SD = 23.1 years). A total of 350 strains were isolated from corneal scrapings and were all culture-positive. Species identification demonstrated that *S. pneumoniae* was the most frequently isolated organism, accounting for 29.8% (n = 108) of cases. This was followed by *S. mitis* at 24.9% (n = 90) and *S. oralis* at 23.5% (n = 85), collectively representing 78.0% (n = 273) of all *Streptococcal* isolates identified in this cohort. In terms of treatment outcomes, it was found that 87.7% (n = 307) of patients received antibiotic therapy alone, while 12.3% (n = 43) required surgical intervention to address the infection. These findings underscore the importance of timely diagnosis and appropriate management strategies in the treatment of *Streptococcal* keratitis.

The most prevalent risk factor identified was a history of previous ocular surface disease, which accounted for 24.6% of cases. Among these, the most common conditions included glaucoma (6.0%), uveitis (5.1%), blepharitis (3.2%), and scleritis (2.0%). In addition to ocular surface diseases, previous ocular surgery was reported in 21.4% of patients, while ocular trauma was noted in 16.3%, both of which are critical risk factors for the development of keratitis. Systemic diseases were present in 9.1% of the cohort, and co-infection with other pathogens was observed in 8.3% of cases. Furthermore, recent corticosteroid use was identified in 4.0% of patients, indicating a potential immunosuppressive risk. Timeliness of medical intervention varied among patients; 43.1% sought care at an outpatient clinic within two weeks of the onset of symptoms, while 27.1% delayed seeking medical attention for one month or more. These findings emphasize the importance of recognizing predisposing conditions and the need for prompt treatment to mitigate the risk of developing *Streptococcal* keratitis.

### 2.2. Clinical Characteristics of Streptococcal Keratitis Subtypes

Patients with *Streptococcal* keratitis commonly presented with a constellation of clinical features, including corneal ulceration (99.0% of cases), dense infiltration (94.0%), corneal thinning (55.0%), edema (54.0%), and early-stage hypopyon (63.0%). Other notable findings were neovascularization and Descemet’s membrane folds (Figure 1). This distinct clinical presentation can aid in the prompt recognition and management of this sight-threatening corneal infection.

As shown in Table 2, significant differences were observed in the clinical presentation of keratitis caused by various *Streptococcal* species, specifically with respect to the location (*p* = 0.038) and size (*p* = 0.037) of dense infiltrations, as well as the presence of hypopyon (*p* = 0.006) among cases attributed to *S. pneumoniae*, *S. oralis*, and *S. mitis*. Patients with *S. pneumoniae* keratitis were more likely to have central corneal involvement (59.0%) compared to *S. oralis* (26.0%, *p* = 0.011). Small dense infiltrations (<3 mm) were more common in *S. mitis* keratitis (35.5%) than in *S. pneumoniae* (17.9%, *p* = 0.041). Hypopyon was more frequently observed in *S. pneumoniae* keratitis (82.0%) compared to *S. oralis* (47.0%, *p* = 0.002) and *S. mitis* (55.0%, *p* = 0.014). There was no statistically significant difference in the time from symptom onset to first clinical visit among the three *Streptococcal* species. These findings highlight the importance of identifying specific *Streptococcal* species, as they may inform clinical management and treatment strategies.

### 2.3. In Vivo Confocal Microscopy Findings in Streptococcal Keratitis Progression

In the mild stage of *Streptococcal* keratitis, IVCM revealed inflammatory cells confined to the epithelial layer, along with dendriform cells near the basement membrane (Appendix A). As the disease progressed, enlarged dense infiltration, hypopyon, and Descemet’s membrane folds emerged (Appendix A). IVCM showed inflammatory cells in the epithelium and anterior stroma, with dendriform cells near the basement membrane (Appendix A). In the severe stage, large dense infiltration, neovascularization, and extensive hypopyon were observed (Appendix A). IVCM detected inflammatory cells from the epithelium to the posterior stroma, with loss of normal keratocyte structure (Appendix A). These IVCM findings provide detailed insights into the cellular changes associated with the progression of *Streptococcal* keratitis, which may guide clinical management.

Progression of the disease was marked by increased densities of inflammatory and dendritiform cells. Inflammatory cell density rose from 1049 ± 142 cells/mm^2^ in the mild stage to 3207 ± 436 cells/mm^2^ in the severe stage (Appendix A), while dendritiform cell density increased from 238 ± 37 cells/mm^2^ to 657 ± 78 cells/mm^2^ (Appendix A). There were 23 eyes (50.0%) presenting honeycomb-shaped inflammatory cells, 44 eyes (95.7%) inflammatory cell infiltration in nonspecific distribution, 30 eyes (65.2%) normal keratocytes, 40 eyes (87.0%) activated keratocytes with nuclei, and 29 eyes (63.0%) activated keratocytes without nuclei. In addition, dendritiform cells could be seen in 34 eyes (73.9%), epithelial bullae were presented in 25 eyes (54.3%), stromal bullae appeared in 23 eyes (50.0%), “spindles” were seen in 30 eyes (65.2%), and scar tissue could be found in 17 eyes (37.0%).

### 2.4. Antibiotic Resistance Profiles of Streptococcus Species

A total of 277 *Streptococcus* isolates underwent antimicrobial susceptibility testing and were included in this part of the study (Appendix A). The resistant rates of vancomycin, amikacin, gentamicin, ciprofloxacin, ofloxacin, rifampin, levofloxacin, tobramycin, ceftazidime, moxifloxacin, and benzalkonium chloride were evaluated separately.

Subsequently, we conducted an analysis to trace the trends of resistance rates of *S. pneumoniae*, *S. oralis*, and *S. mitis*. The resistance rate for rifamycin (0–14.2%) and glycopeptides (0–7.1%) among *Streptococcus* species remained notably low, with no resistance rate observed over the past decade (Figure 2A). In contrast, the resistance rate for aminoglycosides increased significantly, rising from 64.9% in 2006 to 91.6% in 2014. Furthermore, from 2015 to 2023, there was a marked increase in the resistance rates of fluoroquinolones, which rose from 12.0% to 41.6%, as well as β-lactam antibiotics, which increased from 48.2% to 66.7%.

The antibiotic resistance rates among the three *Streptococcus* species were analyzed (Figure 2B). Amikacin and tobramycin exhibited high resistance rates across all three types of *Streptococcal* keratitis. Specifically, *S. pneumoniae*, *S. oralis*, and *S. mitis* showed resistance rates of 90%, 84.8%, and 69.8% to amikacin and 83.9%, 90.9%, and 79.5% to gentamicin, respectively. Significantly different antibiotic resistance rates among three species were found in amikacin (*p* = 0.010), gentamycin (*p* = 0.007), ciprofloxacin (*p* < 0.001), ofloxacin (*p* = 0.005), levofloxacin (*p* < 0.001), benzalkonium chloride (*p* = 0.003), and ceftazidime (*p* = 0.001). A similar finding was found for ciprofloxacin, occupying 90.8% of *S. pneumoniae* isolates, 28.8% of *S. oralis*, and 23.3% of *S. mitis*. On the other hand, *S. pneumoniae* showed relatively low rates of resistance to ofloxacin (5.8%), levofloxacin (2.1%), and benzalkonium chloride (9.1%), whereas in the other two species, the rates were higher: resistance rates to ofloxacin, levofloxacin, and benzalkonium chloride were 28.8%, 21.2%, and 17.2%, respectively, for *S. oralis*, and 21.9%, 19.2%, and 22.7%, respectively, for *S. mitis*. The resistance rate of ceftazidime of *S. oralis* (68.2%) was markedly higher than *S. pneumoniae* (42.5%) and *S. mitis* (47.9%).

### 2.5. Analysis of Prognostic Correlations

In this study, follow-up records of 74 patients were analyzed, categorizing outcomes as good (clinical remission, Figure 3A) or poor (worsening, surgery, Figure 3B). *S. pneumoniae* keratitis had the worst prognosis, with 65% resulting in poor outcomes, whereas *S. mitis* keratitis showed a more favorable prognosis, with only 26% leading to poor outcomes.

Univariate and multivariate analyses (Figure 3C) revealed significant correlations between demographic and predisposing factors and patient outcomes. Notably, a longer interval from symptom onset to the first visit was associated with an increased likelihood of poor outcomes (*p* < 0.001). Other factors, such as age and sex, did not show significant correlations.

Furthermore, an analysis of slit-lamp photography features of *Streptococcal* keratitis indicated that central corneal ulcers (*p* = 0.042), dense central infiltrations (*p* = 0.043), large-sized infiltrations (*p* = 0.047), and hypopyon (*p* = 0.019) were all linked to worse outcomes (Figure 3D). In addition, the appearance of scar tissue on in vivo confocal microscopy (IVCM) was correlated with poor outcomes (Figure 3E, *p* = 0.037). Inflammatory cell and dendritic cell densities were slightly increased in patients with bad outcomes (Figure 3F,G).

## 3. Discussion

*Streptococcal* keratitis is a rapidly progressing corneal disease characterized by poor prognosis, often leading to corneal scarring, perforation, and permanent vision loss [10,11]. This study examines predisposing factors, clinical findings, antibiotic susceptibility patterns, and treatment outcomes of patients with *Streptococcal* keratitis treated at Beijing Tongren Hospital over an 18-year period, aligning with findings from previous studies in Shanghai [12], South India [13], and Australia [14]. The analysis identified *S. pneumoniae* as the most prevalent species, accounting for 29.8% of total *Streptococcal* isolates, consistent with a prior 5-year study that reported *S. pneumoniae* responsible for 38% of bacterial keratitis cases [15].

*Streptococcal* keratitis is typically an opportunistic infection, often arising from immune defects and various risk factors. While contact lens wearing is a common risk factor for bacterial keratitis in developed countries [16,17], none of the patients in this study reported contact lens use. The most prevalent risk factor identified was previous ocular disease, accounting for 24.6% of cases, including conditions such as glaucoma, uveitis, and blepharitis. This prevalence is comparable to a report from northern California (17.7%), but lower than findings from Iran (44.4%) [18]. These pre-existing ocular conditions can compromise the normal structure and defense mechanisms of the eye, increasing susceptibility to bacterial infections. Additionally, 21.4% of cases had undergone prior eye surgery, and 16.3% had experienced ocular trauma, consistent with results from a study in Singapore [19].

We evaluated slit-lamp images to detail *Streptococcal* keratitis manifestations. Nearly all patients (99.0%) developed corneal ulcers, indicating high epithelial barrier breakdown facilitating *Streptococcal* invasion [20]. Corneal thinning was another key feature, potentially linked to virulence factors [16]. Notably, we compared clinical signs among *S. pneumoniae*, *S. oralis*, and *S. mitis* keratitis. *S. pneumoniae* cases showed higher rates of central dense infiltrates and hypopyon, while *S. oralis*/*S. mitis* cases had more paracentral infiltrates and less hypopyon. *S. mitis* cases also tended to have smaller infiltrates. These differences reflected the species-specificity of disease progression, with *S. pneumoniae* being more virulent, which led to a higher risk of rapid disease progression and thus bad outcomes [17]. These findings provide new perspectives on distinguishing these *Streptococcal* keratitis types, informing future treatment.

IVCM serves as a noninvasive tool for the early diagnosis of fungal and amoebic keratitis by detecting fungal filaments and amoebic capsules in real time. However, the IVCM features of bacterial keratitis, particularly *Streptococcal* keratitis, have not been extensively documented [18,19]. Our study enhances the understanding of IVCM characteristics in *Streptococcal* keratitis, revealing that dendritiform cells were present in 73.9% of cases, with inflammatory cells arranged nonspecifically in almost all cases. Epithelial and stromal bullae were noted in half of the cases, occurring more frequently than in amoebic or fungal keratitis [21]. The high density of dendritiform and inflammatory cells suggests a severe immune response. Initially, inflammatory cells are confined to the epithelial layer, but as the disease progresses, they infiltrate the anterior and then the posterior stromal layers, indicating an increasing depth of inflammation. Additionally, the observation of cellular structure destruction allows for assessment of the damage inflicted by *Streptococcus* on epithelial cells and keratocytes.

Treatment of *Streptococcal* keratitis typically involves topical antimicrobial therapy with systemic antibiotics being commonly used in severe cases, and surgical interventions reserved for cases of resistance, progression despite treatment, or corneal perforation [22]. However, due to the widespread use of commonly used topical antibiotics in ophthalmic treatment, bacterial resistance is increasing in some cases, thus limiting the potential efficacy of such drugs [23,24]. A 20-year follow-up study in Taiwan showed increasing antibiotic resistance in Gram-positive bacteria, including rising azithromycin resistance in *S. pneumoniae* [25,26,27]. However, it is important to recognize that the established breakpoints for topical antibiotics are not well defined, making the interpretation of drug sensitivity data challenging. Topical antibiotics have several advantages over systemic medications, including the ability to deliver high concentrations of antimicrobial agents at the desired site of action and reduced systemic toxicity [23]. Even when pathogens exhibit resbuistance in vitro, the concentration of topical antibiotics at the site of infection often exceeds the MIC; therefore, while in vitro resistance patterns are valuable, the local drug concentration at the site of infection should also be considered when evaluating the efficacy of topical antibiotics, because that concentration may still be sufficient to effectively treat the infection. As illustrated in our study, amikacin and tobramycin had higher resistance rates suggesting that they are not suitable for the treatment of *Streptococcal* keratitis, whereas vancomycin and rifampicin had lower resistance rates. We also found that resistance rates to aminoglycosides, β-lactam antibiotics, and fluoroquinolone antibiotics have increased over the past few years. Among them, the rise in antibiotic resistance rates of fluoroquinolones from 12.0% to 41.6% is the most important to note. Fluoroquinolones are often used as the first-line treatment for eye infections, and the medications are generally effective [28,29]. This change highlights new challenges in managing *Streptococcal* keratitis and underscores the need to explore alternative antibiotic options to combat the growing resistance. The varying resistance rates among the three species of streptococci highlight the need for tailored antibiotic therapy in treating *Streptococcal* keratitis. Notably, ciprofloxacin showed significantly higher resistance in *S. pneumoniae* compared to the other species, while *S. oralis* exhibited a notably higher resistance to ceftazidime. Consequently, the use of both ciprofloxacin and ceftazidime would be contraindicated in these cases. However, it is crucial to consider that breakpoints for topical antibiotics are not well defined, and local drug concentrations may exceed the MIC, even in cases where resistance is detected. This discrepancy highlights the need for further research on the pharmacokinetics and efficacy of topical antibiotics in treating ocular infections.

We evaluated clinical signs associated with predicting the clinical outcome of *Streptococcal* keratitis: central corneal ulcer, central dense infiltrate, large-size dense infiltrate, and hypopyon are linked with bad outcomes. A central bacterial corneal infection is very serious because it can pose a severe threat to vision [30]. In infectious keratitis, the presence of a large infiltrate at the onset of the patient’s disease may be related to the late onset of the disease, the inadequate treatment received, the resistance of the pathogen to treatment, or the immune status of the patient, which ultimately leads to a poor prognosis [31]. Hypopyon is formatted by the lesion of leukocytes from blood vessels and gravitates at the bottom of the anterior chamber [32]. The higher proportion of hypopyon in *S. pneumoniae* keratitis than in the other two kinds could explain the correlation of these clinical features with worse outcomes in *S. pneumoniae* keratitis. The presence of inflammation in the anterior chamber indicates deeper lesions of streptococcal infection. This is supported by the results of our analysis of IVCM photographs. Patients with hypopyon tend to have inflammatory cells in the epithelial and stromal layer, suggesting a more severe progression of the infection and therefore a worse prognosis. The presence of scar tissue in the IVCM photographs also suggests severe progression of the infection and a poor prognosis.

This study has several shortcomings; due to its retrospective nature, only part of the participants can be included in the analysis of clinical features and antibiotic susceptibility patterns. The lack of data from follow-up examinations also limits the results of this study. Despite the shortness, this study is the largest study describing *Streptococcal* keratitis, elucidating the clinical features and risk factors for this disease, as well as its antibiotic susceptibility patterns.

In summary, this study provides a comprehensive analysis of *Streptococcal* keratitis, revealing the main risk factors, such as prior ocular surface disease, previous ocular surgery, and ocular trauma. We also found notable differences in clinical features and antibiotic susceptibility among various *Streptococcal* species. Furthermore, in vivo confocal microscopy (IVCM) proves to be an invaluable tool for monitoring the progression of infection, enhancing our understanding and management of this condition.

## 4. Materials and Methods

### 4.1. Participants

This retrospective study examined all cases of *Streptococcal* keratitis at Beijing Tongren Hospital from 1 January 2006 to 30 December 2023. The study adhered to the Declaration of Helsinki and received approval from the Medical Ethics Committee of Beijing Tongren Hospital (TRECKY2015-KY09). Cases were defined by clinical presentation and laboratory confirmation of *Streptococcal* etiology. Patients with incomplete initial or follow-up records were excluded.

### 4.2. Clinical Diagnosis Procedures

Patients’ information, including demographics, predisposing factors, clinical features, initial diagnosis, treatments, and follow-up records, was extracted from the electronic health record system using a standardized Excel proforma. Predisposing factors for *Streptococcal* keratitis were categorized as trauma, previous ocular surgery, systemic disease, previous ocular surface diseases, co-infection at diagnosis, and recent corticosteroid use.

Upon presentation, all patients underwent visual acuity measurement and slit-lamp biomicroscopy examination. Two experienced ophthalmologists carefully evaluated the clinical signs, including the location and size of the corneal ulcer and infiltrate, corneal thinning, neovascularization, Descemet’s membrane folding, and hypopyon. The size of the corneal ulcer and stromal infiltration was measured using corneal photographs, calculated by multiplying the longest diameter and the longest perpendicular width.

The Rostock Cornea Module of the Heidelberg Retina Tomograph III (HRT III/RCM, Heidelberg Engineering, Heidelberg, Germany) was utilized to obtain in vivo confocal microscopy (IVCM) images covering an area of 400 × 400 µm. IVCM photographs of *Streptococcal* keratitis at various stages of progression were evaluated across different corneal layers. The stages of *Streptococcal* keratitis were classified according to the different degrees of ulcer size [33]. Stage I: Early signs of infection with limited corneal lesions (≤3 mm). Stage II: Moderate infection with visible corneal inflammation, hypopyon and potential thinning with medium-sized lesions. Stage III: Advanced infection with significant corneal involvement, large-sized lesions (≥6 mm), hypopyon, neovascularization, possible perforation, or risk of endophthalmitis. The maximum density of inflammatory and dendritiform cells, from the epithelial to the endothelial layers, was quantified using the Cell Count function of ImageJ software v1.4.3.x [34].

Microbiological investigation was conducted for patients with clinically suspected corneal infections. This involved microscopic examination of corneal scrapings, including Gram staining, followed by microbiological culture and antimicrobial susceptibility testing. Corneal scrapes were inoculated on appropriate media, including chocolate agar, blood agar, and Sabouraud dextrose agar, to culture fastidious organisms, bacteria, and fungi, respectively. Isolate identification, including the detection of any mixed infections, was performed using the MALDI-TOF-MS system (Bruker, Bremen, Germany). All diagnostic procedures were carried out by experienced ophthalmologists and laboratory staff.

### 4.3. Antibiotic Susceptibility Testing

Antimicrobial susceptibility testing was interpreted following the Clinical and Laboratory Standards Institute (CLSI) guidelines. Antibiotic susceptibility, including vancomycin, ceftazidime, ofloxacin, levofloxacin, rifampin, moxifloxacin, ciprofloxacin, benzalkonium chloride, amikacin, gentamicin, and tobramycin, was assessed using the Kirby–Bauer disk diffusion method. The detailed CLSI breakpoints for each antibiotic are presented in Appendix A.

Isolates were prepared to match the 0.5 McFarland standard for turbidity, ensuring a consistent bacterial concentration for testing of approximately 1.5 × 10^8^ CFU/mL. Using a sterile swab, the standardized bacterial suspension was evenly spread across the surface of Mueller–Hinton agar plates to create a uniform lawn using a sterile swab. After allowing the inoculum to settle briefly, antibiotic-impregnated disks (vancomycin, ceftazidime, ofloxacin, levofloxacin, rifampin, moxifloxacin, ciprofloxacin, benzalkonium chloride, amikacin, gentamicin, and tobramycin) were carefully placed onto the agar surface using sterile forceps after allowing the inoculum to settle briefly. The plates were then incubated in an aerobic environment at 35 °C (21% O2) for 18–24 h.

Following incubation, the diameter of each inhibition zone around the antibiotic disks was measured to the nearest millimeter. These measurements were interpreted using CLSI breakpoints to classify each isolate’s response as resistant (R), intermediate (I), or sensitive (S) based on established zone diameter breakpoints for each antibiotic. Additionally, control strains (e.g., *E. coli* ATCC 25922) were used to validate the accuracy of the antimicrobial susceptibility testing and were tested at regular intervals to ensure the consistency of the results. All data on cases with mixed infections are presented in Appendix A.

### 4.4. Treatment and Outcome

The treatment of *Streptococcal* keratitis involves both antibiotic therapy and surgical intervention. Adequate treatment is crucial to prevent serious complications, including empiric subconjunctival antibiotic therapy or systemic antibiotic therapy targeting any scleral or intraocular extension of the infection. Initial recommendations included monotherapy with fluoroquinolone, while current guidelines suggest empiric treatment with fluoroquinolone or intensive combination therapy with cefazolin and gentamicin [35]. The antibiotic regimen is subsequently modified based on clinical response and antibiotic susceptibility testing. Therapeutic penetrating keratoplasty (PK) remains the primary surgical intervention for rapidly progressing infections. The indications for surgical procedures are determined by experienced experts [36].

Prognosis was evaluated by analyzing the patients’ follow-up medical records for at least one month. During follow-up, the clinical progression of the disease was carefully monitored through regular corneal assessments and slit-lamp examinations to evaluate the size and depth of ulcers or infiltrates. The presence of complications such as corneal thinning, perforation, or the need for surgical intervention was noted. Patients were classified as having a good outcome if they experienced relief from the ulcers or infiltrates using only topical antibiotics. This was defined as complete or near-complete resolution of corneal lesions without the need for further surgical intervention, and with no signs of infection recurrence. In contrast, bad outcomes were defined as an inadequate response to antibiotic treatment, characterized by the enlargement of ulcers or infiltrates, progressive corneal thinning, or the need for surgical intervention. If surgical intervention, such as penetrating keratoplasty, was required due to the severity or progression of the infection, it was classified as a poor outcome.

### 4.5. Statistical Analysis

All statistical analyses were performed using SPSS 26.0 (SPSS, Chicago, IL, USA). Descriptive statistics were calculated, and the data were presented in figures and tables. Continuous variables were described using appropriate measures of central tendency and dispersion, while categorical variables were reported as percentages. The Chi-square test or Fisher’s exact test was used to compare the clinical characteristics of patients with *Streptococcal* keratitis and cases with good and bad outcomes. Bonferroni correction was applied to account for multiple comparisons. A *p*-value of 0.05 or less was considered statistically significant.

## 5. Conclusions

This study offers a detailed analysis of *Streptococcal* keratitis, highlighting key risk factors, differences in clinical features and antibiotic susceptibility among species, and the valuable role of in vivo confocal microscopy in monitoring infection progression and improving management.

## Figures and Tables

**Figure 1 antibiotics-13-01190-f001:**
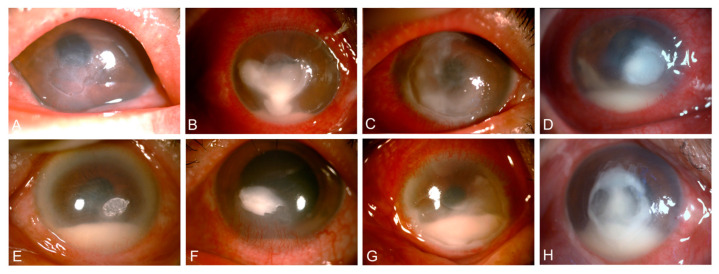
Slit-lamp images showing typical signs of *Streptococcal* keratitis. (**A**) Ulceration, (**B**) dense infiltrate, (**C**) corneal thinning, (**D**) edema, (**E**) Descemet’s membrane folds, (**F**) neovascularization, (**G**) conjunctival hyperemia, and (**H**) hypopyon.

**Figure 2 antibiotics-13-01190-f002:**
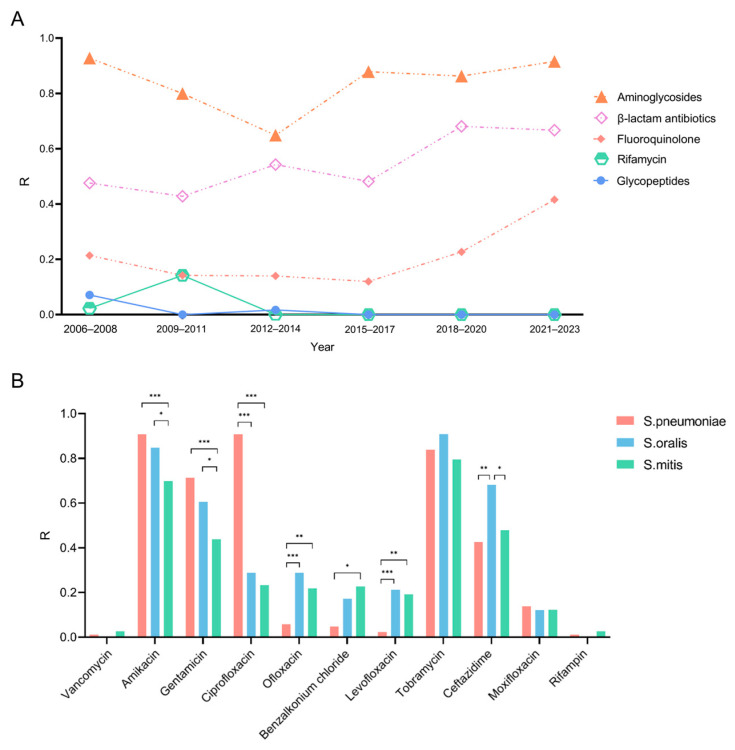
Changes in resistance rates of cornea-derived *Streptococci* over the last 18 years and a comparison of resistance rates among *S. pneumoniae*, *S. oralis*, and *S. mitis*. R: the percentage of isolates resistant to each antibiotic. (**A**) Changes in antimicrobial resistance patterns of cornea-derived *Streptococci* over the past 18 years. (**B**) Comparison of resistance rates among *S. pneumoniae*, *S. oralis*, and *S. mitis*. * = *p*-value < 0.05; ** = *p*-value < 0.01; *** = *p*-value < 0.001.

**Figure 3 antibiotics-13-01190-f003:**
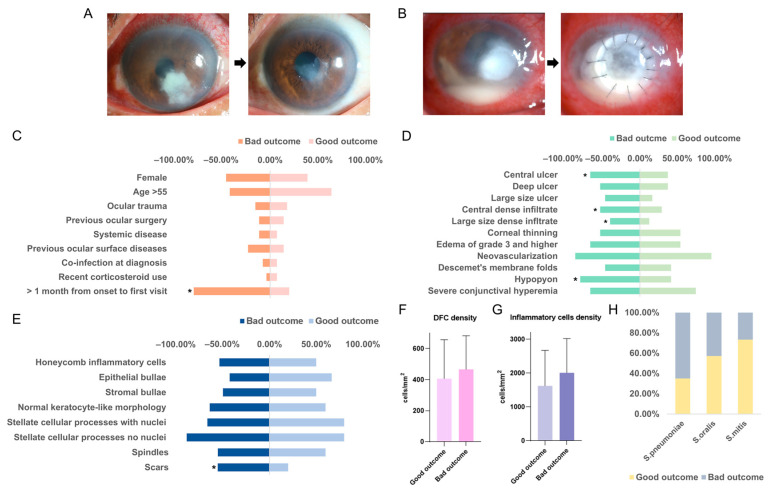
Risk factors may affect outcome and prognostic differences among *Streptococcus* pneumoniae *Streptococcus* oralis and *Streptococcus* mitis keratitis. (**A**) The clinical manifestation of a good outcome of patients with *Streptococcal* keratitis. (**B**) The clinical manifestation of a good outcome of patients with *Streptococcal* keratitis. The correlation between demogsraphic and predisposing data (**C**), initial clinical characteristics (**D**), IVCM cellular features (**E**), dendritiform cell density in IVCM photographs (**F**), inflammatory cells density in IVCM photographs (**G**), and the outcome. (**H**) A comparison of outcomes of *Streptococcus* pneumoniae, *Streptococcus* oralis, and *Streptococcus* mitis keratitis. *: significant *p*-value < 0.05.

**Table 1 antibiotics-13-01190-t001:** Demographic and predisposing factors of *Streptococcus* keratitis.

Parameters	Patient (n = 350)
**Demographics**	
Age (mean ± SD)	49.1 ± 23.1
Female, n (%)	165 (47.1%)
Laterality	
Left eye, n (%)	151 (43.1%)
Right eye, n (%)	193 (55.1%)
Both eyes, n (%)	6 (1.7%)
**Predisposing factors, n (%)**	
Ocular trauma	57 (16.3%)
Previous ocular surgery	75 (21.4%)
Corneal surgery	22 (6.3%)
PK	5 (1.4%)
DSAEK	10 (2.9%)
AMT	5 (1.4%)
Gunderson flap	2(0.6%)
Other surgery	53 (15.1%)
Cataract Surgery	17 (4.9%)
Eyelid plastic and reconstructive surgery	8 (2.3%)
Ahmed glaucoma valve implantation	7 (1.9%)
Excision of eyelid and conjunctival tumors	6 (1.7%)
Vitrectomy	4 (1.1%)
Other surgeries	11 (3.1%)
Systemic disease	32 (9.1%)
Diabetes mellitus	14 (4.0%)
Ankylosing Spondylitis	5 (1.4%)
Sjögren’s Syndrome	3 (0.8%)
Ankylosing Spondylitis	3 (0.8%)
Other diseases	8 (2.3%)
Previous ocular diseases	86 (24.6%)
Glaucoma	21 (6.0%)
Uveitis	18 (5.1%)
Blepharitis	11 (3.1%)
Scleritis	7 (1.9%)
Other diseases	29 (8.3%)
Co-infection at diagnosis	29(8.3%)
Other bacteria	16(4.6%)
Fungi	11(3.1%)
Amoeba	2(0.6%)
Recent corticosteroid use	14 (4.0%)
**Onset time, n (%)**	
<2 weeks	151 (43.1%)
2 weeks to 1 month	104 (29.7%)
>1 month	95 (27.1%)
**Species of *Streptococcus* isolated, n (%)**	
*S. pneumoniae*	108 (29.8%)
*S. mitis*	90 (24.9%)
*S. oralis*	85 (23.5%)
*S. sanguis*	23 (6.4%)
*S. constellatus*	7 (1.9%)
*S. salivarius*	6 (1.7%)
*S. agalactiae*	6 (1.7%)
*S. dysgalactie*	6 (1.7%)
*S. goronii*	5 (1.4%)
*S. anginosus*	4 (1.1%)
*S. pyogenes*	3 (0.8%)
Others	7 (1.9%)
**Treatment, n(%)**	
Antibiotic medications	307 (87.7%)
Antibiotic medications and surgery	43 (12.3%)

Note: n = number of patients. PK = Penetrating Keratoplasty. DSAEK = Descemet’s Stripping Automated Endothelial Keratoplasty. AMT = Amniotic Membrane Transplantation.

**Table 2 antibiotics-13-01190-t002:** Comparison of clinical signs of *S. pneumoniae*, *S. oralis*, and *S. mitis* keratitis.

Clinical Signs	*S. pneumoniae* (n = 39)	*S. oralis* (n = 30)	*S. mitis* (n = 31)	*p*-Value
**Location of ulcer**				0.358
Central	25	15	14	
Paracentral	11	12	15	
Peripheral	3	1	1	
None	0	0	1	
**Size of ulcer**				0.555
<3 mm	5	4	7	
3–6 mm	27	19	21	
>6 mm	7	7	3	
**Location of dense infiltrate**				0.038 *
Central	23	8	11	
Paracentral	14	12	15	
Peripheral	2	4	3	
Complete cornea	0	1	1	
None	0	5	1	
**Size of dense infiltrate**				0.037 *
<3 mm	7	6	11	
3–6 mm	28	19	13	
>6 mm	4	5	7	
**Corneal thinning**				0.798
Yes	22	15	18	
No	17	15	13	
**Neovascularization**				0.300
Yes	36	28	31	
No	3	2	0	
**Descemet’s membrane folds**				0.337
Yes	13	12	7	
No	26	18	24	
**Hypopyon**				0.006 *
Yes	32	14	17	
No	7	16	14	
**Conjunctival hyperemia**				0.370
1	0	3	2	
2	6	6	5	
3	33	21	24	

Note: *p*-values were calculated using Chi-square test and Fisher exact test. * = significant *p*-value < 0.05.

## Data Availability

The information used in our study is available upon request from the corresponding author. The dataset is not available to the public due to the need to protect patient privacy.

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
