# Peer review of "Risk Factors, Clinical Characteristics, and Antibiotic Susceptibility Patterns of Streptococcal Keratitis: An 18-Year Retrospective Study from a Tertiary Hospital in China"

_antibiotics, 2024, doi:10.3390/antibiotics13121190_

Round 1

Reviewer 1 Report

Comments and Suggestions for Authors

Dear authors, the topic of Streptococcal Keratitis is of interest. But the microbiologic diagnosis is incorrect and incompletely presented.

Please specify:

1. what is the antimicrobial susceptibility testing interpretation standard used. 

2. why the susceptibility of S. pneumoniae and viridans group streptococci to aminoglycosides, ceftazidime, ciprofloxacin has been tested, as none of the standards recommend this

3. explain or correct the name of the antibiotic "benzazolin"

4. the chapter Antibiotic susceptibility testing needs to be rewritten; it is insufficiently detailed

Reviewer 2 Report

Comments and Suggestions for Authors

Material and Methods must be placed before Results and Discussions; please modify

line 57 - age ranging from 2 months (...) - this could be a birth infection; please share your experience with such keratitis + literature

line 58-59 - (...) infected cornea scraping (...) all were from the cornea - unclear phrasing; please modify 

Table 1, Predisposing factors - previous ocular surgery, other surgery, systemic disease, previous ocular diseases; please be more specific (e.g. pterygium, cataract surgery, diabetes)

Table 1, Complications - glaucoma - it is almost impossible to assess the optic nerve and visual field in bacterial keratitis; secondary ocular hypertension is more appropriate 

Table 1, Complications - corticosteroid use and co-infection at diagnosis are not complications of keratitis, rather risk factors; please modify

Table 1, Treatment - the authors state in the text that 43 patients required surgery besides antibiotics, but from the table results that 43 patients underwent surgery alone; please modify

Line 69-71 - glaucoma and uveitis are stated as ocular surface risk factors, which they are not; please modify

Figure 2, (S) and (T) - there is no definition of the "X" axis, although the text suggests that it is progression of keratitis; please define

Line 139 - 226 Streptococcal isolates (...); why only 226 from 350? Please comment

Line 140-142 - some strains of Streptococcus are still sensible to penicillin; please comment why this drug was not included in your study

Line 209-210 - same as Line 69-71

Line 250 - (...) beta-lactam antibodies (...); they are beta-lactam antibiotics; please modify 

Line 271 - (...) with more bad (...); worse would be a more suitable rephrase for more bad; please modify

Line 271-272 - please explain the term "inferior chamber" in this context

Line 333 - Treatment and Outcome - whole paragraph is unclear and on the short side. General antibiotic is recommended from presentation? Monotherapy with fluoroquinolone is topical?  Was there no treatment for fungal/acanthamoeba co-infection? Therapeutic penetrating keratoplasty is not the only surgical option for these cases; please discuss other options in short and long term (e.g. tectonic PK, DALK, Gunderson flap, evisceration). Do steroids play any role in infectious keratitis in your Clinic? Is there any protocol in your Clinic regarding the patient with infectious keratitis? If not, please modify this paragraph explaining step by step the whole process of diagnostic, treatment, and follow up that happens to the patient with infectious keratitis in your Clinic.

Line 344-349 - please comment why visual acuity is not among the prognostic factors.

Comments on the Quality of English Language

- Medical English language must be improved (e.g. terms like good and bad when referred to prognostic can be changed to favorable/improve or unfavorable/poor)

- Non-medical English must also be improved (there is no place for terms like more bad at this academic level)

Reviewer 3 Report

Comments and Suggestions for Authors

This manuscript describes the clinical features, risk factors and antimicrobial susceptibility/resistance results for species of Streptococcus associated with keratitis.  Most of my review focuses on the antimicrobial susceptibility results and I leave the main ophthalmology findings to others to comment on.  The following are a few points which I believe will enhance the presentation of the data.

1) in the abstract it is stated susceptibility of streptococcal keratitis.  This wording is awkward.  susceptibility or resistance is measured for microorganisms and antibiotics and as such the way its is written is incorrect

2) organism names in the manuscript should be in italics

3) in section 2.1, lines 58-59 do not read correctly

4)Figure 3, the scale on the y axis is unclear to me in terms of what is actually being shown.

5) regarding the stats pertaining to antimicrobial susceptibility/resistance, what is actually being reported.  is it the trend in antimicrobial susceptivity/resistance over time or something different.  The statistical presentation of the data on susceptibility is very unclear.

6)under methods, the section describing antimicrobial susceptibility testing is very thin and incomplete.  some additional points to consider are CFU/ml, what atmosphere was incubation done in (O2 or CO2), what breakpoints were used (CLSI, EUCAST), were control strains used and how often were they tested.

7) a table showing the antimicrobial susceptibility results (as number or percent susceptible/resistant) would be helpful (at least for the most recently collected isolates.

8) there was was no mention how the organisms were identified and were any  found in mixed infections

9) was susceptibility determined for the less commonly recovered Streptococcal strains as there was no further mention of these.

10) no comment is made on systemic versus topical antimicrobial therapy and  how might the susceptibility/resistance data be interpreted.  Breakpoints do not exist for topical antibiotics.  Drug concentrations delivered in topical treatments often (but not always) exceed MICs, even if the organism tests resistant.

Round 2

Reviewer 1 Report

Comments and Suggestions for Authors

Thank you for the clarification.

The paper can be published in its current form.

Reviewer 2 Report

Comments and Suggestions for Authors The article has been sufficiently improved to be published in Antibiotics.

Comments on the Quality of English Language

/